# OpenReview forum: "Toward Safer Diffusion Language Models: Discovery and Mitigation of Priming Vulnerability"
_ICLR.cc/2026/Conference — ICLR 2026 Poster_

### Official Review · Reviewer_GBUt · 2025-10-20

**Soundness:** 3
**Presentation:** 2
**Contribution:** 3
**Rating:** 6
**Confidence:** 3

**Summary:**

This paper investigate the  priming vulnerability in diffusion language models , which affirmative tokens could be used during intermediate denoising steps to achieve attacks.. To address this, the authors propose Recovery Alignment which teach DLMs to recover safe responses from contaminated intermediate states. Experiments show the proposed methods can substantially reduces jailbreak  attack success rates.

**Strengths:**

1. This paper provides the systematic safety analysis of diffusion-based language models and highlights a vulnerability unique to their parallel denoising mechanism.
2. The proposed alignment performance is strong which substantially reduces jailbreak  attack success rates.
3. The evaluation is comprehensive which covers multiple models and datasets.

**Weaknesses:**

1. The LLM usage is not described in the paper.  The paper frequently contains long compound sentences composed of multiple short sentences, which is a typical LLM type writing.
2. Despite the differences in form, the priming vulnerability is discussed by several prior works regarding ARM[1,2], the author should mentioned these work in the paper and discuss the difference.
3. The presentation need to be improved. Some sentences are confusing such as"Moreover, they do not discuss how a more realistic attacker, who cannot intervene in the denoising process, could exploit this vulnerability. In this work, we design an attack that intervenes in the denoising process for comprehensive evaluation. " Even though with the following sentences i understand what author what to say, but this is confusing.
4. The ablation is not comprehensive. The author should conduct counterfactual experiments (e.g., replacing non-affirmative tokens or injecting random noise tokens) to verify whether the safety failures specifically depend on token semantics. Meanwhile, the sampling parameters already prove the play a import role in jailbreak success rate[3],  it would be good if related ablation is included.
5. Theorem 4.1 assume a monotonicity condition which is overly strong, if this condition satisfied that means that as denoising proceeds, the likelihood of generating the target response r should never decrease. In diffusion model, each step involves stochastic re-masking and resampling such the monotonicity condition is impossible.  The authors should thoroughly revise the theoretical part.



[1]Huang, Yuyi, et al. "Intrinsic Model Weaknesses: How Priming Attacks Unveil Vulnerabilities in Large Language Models." arXiv preprint arXiv:2502.16491 (2025).

[2]Miao, Ziqi, et al. "Response attack: Exploiting contextual priming to jailbreak large language models." arXiv preprint arXiv:2507.05248 (2025).

[3]Huang, Yangsibo, Samyak Gupta, Mengzhou Xia, Kai Li, and Danqi Chen. "Catastrophic jailbreak of open-source llms via exploiting generation

**Questions:**

1. Does recovery alignment need to manually crafted the  contaminated intermediate states? If so, Would this increase the training cost and make it difficult to scale?

Others see weakness.

---

> ### Author Response · Authors · 2025-11-21
> **Response (1/2)**
>
> **Thank you very much for taking the time to review our paper.** We deeply appreciate your encouraging evaluation of our work and the valuable suggestions you have shared. We **uploaded a revised version** of the paper, and all changes are highlighted in **red.**
>
> ---
>
> # W1. LLM usage
>
> We apologize for not providing a clear description of our LLM usage. We primarily used LLMs for writing assistance, and we have added this clarification to Section 8.
>
> ---
>
> # W2. Priming vulnerability in ARMs
>
> We sincerely appreciate the reviewer pointing out these important related studies
>
> We believe that **the primary difference between [A, B] and our work lies in where the harmful context resides.** [A, B] focus on the priming vulnerability at the **input level**, showing that harmful responses can be triggered when malicious context is included in the input, such as the prompt [A] or the dialogue history [B]. In contrast, our work focuses on the priming vulnerability **in the denoising process that is specific to MDLMs.** Specifically, we analyze how each token in a single output sequence influences the subsequent denoising process.
>
> To clarify these differences, we have revised the Related Works section and Appendix B.2 to explicitly cite [A, B]. We believe these revisions make the positioning and contributions of our work clearer. Once again, we are grateful for this valuable suggestion.
>
> [A] Huang, Yuyi, et al. "Intrinsic Model Weaknesses: How Priming Attacks Unveil Vulnerabilities in Large Language Models." arXiv preprint arXiv:2502.16491 (2025).
>
> [B] Miao, Ziqi, et al. "Response attack: Exploiting contextual priming to jailbreak large language models." arXiv preprint arXiv:2507.05248 (2025).
>
> ---
>
> # W3. **Improving Explanations**
>
> Thank you for pointing this out. We have revised the indicated sentences to make them clearer. In the camera-ready version, we will review all sentences again and improve the clarity throughout the paper.
>
> ---
>
> # W4. Additional Ablation
>
> Thank you for the helpful suggestions. **We have strengthened the experiments in Appendix C.1 on how the semantics of injected tokens affect safety failures.** Below, we summarize the experimental results.
>
> To dissect the effect of the inserted token, we consider three types of anchors:
>
> - **Affirmative tokens** include tokens such as “Yes”, “Sure”, and “Finally”, which explicitly suggest that the model is about to provide or continue an answer.
> - **Harmful tokens** consist of tokens such as “weapon” and “bomb” that are semantically aligned with the malicious intent of the query.
> - **Neutral tokens** include tokens such as “football” and “coffee”, which are semantically unrelated to the query and do not carry obvious safety connotations.
>
> Results (see Figure 4 in the revised version) show that **affirmative tokens reduce the refusal probability the most**. At the same time, **some neutral tokens such as “mountain” also decrease the refusal probability**. We speculate that this is because such words do not appear in typical refusal phrases during training and thus create an out-of-distribution pattern. In contrast, **harmful tokens do not consistently lower the refusal probability**, likely because they frequently appear within standard refusal justifications, and therefore remain compatible with high refusal probability.
>
> We also evaluate how sampling parameters influence the jailbreak success rate using the anchoring attack (Appendix C.5).
>
> - For the unaligned original model, increasing the generation length raises the ASR even without an attack.
> - For the RA-aligned model, shorter generation lengths lead to higher ASR with fewer inserted tokens. This is likely because, when the generation is short, the inserted tokens dominate the context, making it harder for the model to produce a safe response naturally.
>
> We plan to include these experiments in the main text of the camera-ready version if the page limit allows. Thank you again for your helpful suggestion.

---

> > ### Author Response · Authors · 2025-11-21
> > **Response (2/2)**
> >
> > # W5. Monotonicity assumption
> >
> > Thank you for raising this very important point. Following your suggestion, we added further explanations of the monotonicity assumption in Section 4.2 and Appendix C.1. Our interpretation is as follows:
> >
> > First, the monotonicity assumption in Theorem 1 is not intended as a strict or universal rule. We treat it as a **practical and empirically reasonable behavior that naturally appears in how current MDLMs perform inference**. In typical implementations, once an unmasked token is sampled, it is essentially fixed in all subsequent steps. As a result, the model performs inference with an increasingly informative context about the output sequence $r$ as the process progresses. Consequently, for a harmful response $r$, its probability under the output distribution is expected to be more diffuse in the early steps, where many candidate continuations remain possible, and more concentrated in the later steps, where consistency with already fixed tokens substantially narrows the set of plausible candidates.
> >
> > **We do not claim that this property holds uniformly for every possible response**. For instance, highly unnatural strings, such as extreme repetitions of a particular word, may violate the assumption. However, our focus is on natural harmful content that the model could realistically generate. Such responses receive non-negligible probability mass during pre-training or fine-tuning, making the assumption more likely to hold in these cases.
> >
> > To support our claim, **Appendix C.1 empirically evaluates the assumption using diverse harmful prompts and responses from the JBB-Behavior dataset.** We analyze how the output distribution changes as more injected tokens are added. The results show a clear monotonic trend. While we acknowledge that the assumption is an empirical approximation rather than an absolute rule, these results provide evidence that the assumption is appropriate for harmful responses in practice.
> >
> > ---
> >
> > # Q1. Generation of Contaminated Intermediate States
> >
> > Thank you for your insightful question. Recovery Alignment does **not** require manually crafted contaminated intermediate states. We automatically create them by randomly selecting partial tokens from a set of pre-collected harmful responses. Because of this, we do not see a significant increase in training cost, and we believe the method can scale to larger settings.
> >
> > ---
> >
> > ### **Thank you again for taking the time to review our paper.**
> >
> > We are also truly grateful for your favorable evaluation of our work.

---

> > > ### Comment · Reviewer_GBUt · 2025-11-25
> > > **Thanks for clarification**
> > >
> > > Thanks to the authors for the clarifications and for taking the time to respond. I mentioned the use of LLMs in my initial review because this was a requirement for ICLR this year; I appreciate the authors’ work and did not want the paper to risk a desk rejection for this reason.
> > >
> > > The empirical validation of the monotonicity assumption and the ablation studies have addressed my concerns. My suggestion was intended only to encourage strengthening the rigor of the theoretical.
> > >
> > > I will maintain my score now, and if necessary, I will consider raising it.

---

> > > > ### Author Response · Authors · 2025-11-27
> > > > **Thank you for your response**
> > > >
> > > > Thank you very much for your follow-up and clarifications.
> > > >
> > > > We sincerely appreciate your constructive feedback and positive evaluation of our work.

---

### Official Review · Reviewer_ztRH · 2025-10-31

**Soundness:** 2
**Presentation:** 3
**Contribution:** 3
**Rating:** 6
**Confidence:** 3

**Summary:**

The authors investigate safety vulnerabilities in Masked Diffusion Language Models (MDLMs), a class of parallel, iterative denoising-based text generators. They identify a new risk priming vulnerability: inserting or generating affirmative tokens during intermediate denoising steps can steer an otherwise aligned MDLM toward producing harmful content. To mitigate this issue, the authors design Recovery Alignment (RA) trains MDLMs to “recover” safe responses via RLHF-style optimization. Experiments on LLaDA, LLaDA 1.5, and MMaDA models show that RA significantly reduces attack success rates while maintaining general performance.

**Strengths:**

1. The authors study an interesting and important problem given the fast-growing of DLMs.
2. The identified risk is unique to DLM and is interesting.
3. The proposed attack has high ASR. The proposed alignment method is empirically useful.
4. The paper provides a theoretical analysis of the identified risk.

**Weaknesses:**

1. Unclear definition of "affirmative tokens" and it is unknown how the authors find all such tokens in the model full vocabulary.
2. The theory proposed lacks empirical validation.
3. The experiments are not conducted on larger DLMs, lacking experiments on generalization.

**Questions:**

See weakness.

---

> ### Author Response · Authors · 2025-11-21
> **Response**
>
> **Thank you very much for taking the time to review our paper.** We sincerely appreciate your favorable assessment of our paper, as well as your constructive and thoughtful feedback. We **uploaded a revised version** of the paper, and all changes are highlighted in **red.**
>
> ---
>
> # W1. Clarification on the definition and identification of “affirmative tokens”
>
> Thank you for pointing out that our definition of *affirmative tokens* and their identification procedure was not sufficiently clear.
>
> We use *affirmative tokens* to denote tokens in the response that **endorse or advance a harmful intent.** Specifically, once such tokens appear at an intermediate step of the denoising process, subsequent generation tends to be steered toward a harmful response. However, which tokens act as strong “affirmative” signals depends on the training data, safety alignment, and the particular MDLM. Therefore, we do **not** claim that there is a fixed, model-independent lexicon of affirmative tokens, and we do **not** find all such tokens in the full vocabulary.
>
> In our anchoring attack, we simply sample tokens from the harmful response and inject them as candidate affirmative tokens. Moreover, Appendix C.1 provides a detailed analysis of which individual tokens most strongly alter the probability of producing a refusal. Our results show that overtly harmful words such as *“kill”* and *“bomb”* have relatively limited influence, whereas discourse markers like *“Finally”* and *“Next”*, which implicitly signal progress toward an explanation, reduce the refusal probability substantially.
>
> ---
>
> # W2. Lack of empirical validation for the theorem
>
> As the reviewer points out, we do not run a dedicated experiment that directly verifies the theorem; however, **we empirically validate both its key assumption and the practical effectiveness of the attack that is derived from it**:
>
> - In Appendix C.2, we empirically assess the empirical validity of this assumption using a wide range of prompts and harmful responses $r$ on the JBB-Behaviors dataset. We observe that, as the number of injected harmful tokens increases, the generation probability of $r$ increases accordingly, supporting the monotonicity assumption used in the theorem.
> - Theorem 4.1 directly motivates using the first-step log-likelihood as a surrogate objective for GCG. Consistent with this theoretical insight, Table 1 shows that the resulting First-Step GCG achieves substantially higher ASR and lower computational cost than Monte Carlo GCG.
>
> Thus, while we fully agree with the reviewer that empirical validation of theoretical analysis is important, we respectfully believe that these results provide concrete empirical support for our theorem.
>
> ---
>
> # W3. The experiments are not conducted on larger MDLMs
>
> We agree that evaluating larger MDLMs would be valuable for assessing the generalization of our findings. However, to the best of our knowledge, **there were no publicly available open-source MDLMs larger than the LLaDA 8B models we used.** In principle, our proposed methods are directly applicable to larger MDLMs once such models become available. Accordingly, while we acknowledge this limitation, we leave experiments on larger MDLMs as an important direction for future work.
>
> ---
>
> ### **Thank you again for taking the time to review our paper.**
>
> We are also truly grateful for your favorable evaluation of our work.

---

### Official Review · Reviewer_zsC5 · 2025-11-01

**Soundness:** 3
**Presentation:** 3
**Contribution:** 3
**Rating:** 8
**Confidence:** 2

**Summary:**

This paper focuses on the robustness of DLM, discovers and tries to mitigate the priming vulnerability of such models. Priming vulnerability, detailedly discussed in Section 4, refers to a jailbroken behavior of the DLMs that when affirmative tokens appear at an intermediate step of denoising, subsequent generation would have greater probability of producing harmful response. Experiments with two threat models are presented, which demonstrate clear evidence of the priming vulnerability. To mitigate this vulnerability, this paper proposes the recovery alignment. According to Tables 2-4, RA exhibits clear advantage comparing to exsiting alignments.

**Strengths:**

1. (Novelty and soundness) This paper focus on the robustness of diffusion models, which is an important topic that is left less studied. This paper not only discover the safety risk of DLM (called the priming vulnerability), but also propose to mitigate this risk by proposing recovery alignment. I believe this paper would help improve the overall robustness of DLMs.
2. The experimental results are significant.

**Weaknesses:**

1. I cannot find a discussion on the related works on the metrics of jailbreak attacks (i.e., how to judge whether a model's response is
controversial). Since this is directly related to the final ASR and is rapidly developing overtime, I suggest include a brief discussion on the metrics, in addition to the methods.

**Questions:**

See the weakness part.

---

> ### Author Response · Authors · 2025-11-21
> **Response**
>
> **Thank you very much for taking the time to review our paper.** We are also grateful for your positive evaluation of our work and for the insightful comments you provided. We **uploaded a revised version** of the paper, and all changes are highlighted in **red.**
>
> ## Related works on the metrics of jailbreak attacks
>
> Thank you very much for the excellent suggestion. We agree that discussing related work on the metrics of jailbreak attacks would be valuable, and we have added this discussion to the Related Work section.
>
> ### **Thank you again for taking the time to review our paper.**
>
> We are also truly grateful for your favorable evaluation of our work.

---

### Official Review · Reviewer_hNjZ · 2025-11-01

**Soundness:** 3
**Presentation:** 3
**Contribution:** 3
**Rating:** 8
**Confidence:** 2

**Summary:**

This paper identifies and rigorously analyzes a novel security vulnerability, termed "priming vulnerability," in Diffusion Language Models (DLMs). This vulnerability stems from the iterative, parallel denoising process, where the presence of an affirmative token in an intermediate state can steer the subsequent generation toward a harmful response, bypassing safety guardrails.
The authors propose two main contributions:

They introduce the Anchoring Attack (a hypothetical intervention) to characterize the vulnerability and the First-Step GCG (a non-interventional, realistic attack) to exploit it efficiently.

They propose Recovery Alignment, a DLM-specific safety alignment method that trains the model to recover from adversarially "contaminated" intermediate states back to a safe response, utilizing an RLHF-style objective with a linear curriculum.

**Strengths:**

Although I am not familiar with the DLMs, I believe there are still many Pros:
* **Novel and Timely Focus**: The work is highly relevant and timely, focusing on the unique safety challenges of an emerging model class (DLMs) whose non-causal inference fundamentally differs from Autoregressive Models. This addresses a critical gap in current LLM safety research.
* **Rigorous Vulnerability Analysis**: The paper provides compelling, quantitative evidence for the "priming vulnerability.", including:
    * Anchoring Attack: It demonstrates that even a minimal intervention at t =1 can significantly increase the Attack Success Rate (ASR increases from 2% to 21% on LLaDA Instruct). This clearly demonstrates the sensitivity of the intermediate states.
    * First-Step GCG: The authors cleverly leverage the vulnerability to derive a tractable lower bound on the intractable GCG objective, resulting in an attack that is ∼20× faster and significantly more effective than Monte Carlo-based GCG. This provides a strong, efficient evaluation benchmark.
* Well-Motivated Defense: Recovery Alignment directly addresses the root cause of the vulnerability: the failure of traditional alignment to account for contaminated intermediate states. The use of a linear schedule for the intervention step is a sound curriculum learning approach for stabilization.
* Practicality of Defense: The RA framework is instantiated in an RLHF-style that leverages existing harmful query datasets and reward models, suggesting a practical and scalable solution without requiring bespoke data construction.

**Weaknesses:**

* The theoretical justification in 4.1 for this monotonic increase in the log-likelihood of the target response r over the denoising steps needs to be more robustly discussed, especially for different harmful responses r. Does this assumption hold true when r is a diverse set of harmful responses?
* Complexity of RA Training: Although RA is practical in terms of data, the training process involves generating intermediate states, running the partial denoising process, and then using a potentially expensive reward model to score the output.
* Generality of Priming Token: The paper focuses on "affirmative tokens." It would be valuable to discuss if other types of tokens (e.g., specific harmful jargon, boundary tokens) could similarly "prime" the model, or if the mechanism is uniquely linked to affirmation.

**Questions:**

NA

---

> ### Author Response · Authors · 2025-11-21
> **Response**
>
> **Thank you very much for taking the time to review our paper.** We also appreciate your positive assessment of our work and your thoughtful comments. We **uploaded a revised version** of the paper, and all changes are highlighted in **red.**
>
> ---
>
> # W1. Monotonicity assumption
>
> Thank you for raising this very important point. Following your suggestion, we added further explanations of the monotonicity assumption in Section 4.2 and Appendix C.1. Our interpretation is as follows:
>
> First, the monotonicity assumption in Theorem 1 is not intended as a strict or universal rule. We treat it as a **practical and empirically reasonable behavior that naturally appears in how current MDLMs perform inference**. In typical implementations, once an unmasked token is sampled, it is essentially fixed in all subsequent steps. As a result, the model performs inference with an increasingly informative context about the output sequence $r$ as the process progresses. Consequently, for a harmful response $r$, its probability under the output distribution is expected to be more diffuse in the early steps, where many candidate continuations remain possible, and more concentrated in the later steps, where consistency with already fixed tokens substantially narrows the set of plausible candidates.
>
> **We do not claim that this property holds uniformly for every possible response $r$**. For instance, highly unnatural strings, such as extreme repetitions of a particular word, may violate the assumption. However, our focus is on natural harmful content that the model could realistically generate. Such responses receive non-negligible probability mass during pre-training or fine-tuning, making the assumption more likely to hold in these cases.
>
> To support our claim, **Appendix C.1 empirically evaluates the assumption using diverse harmful prompts and responses from the JBB-Behavior dataset.** We analyze how the output distribution changes as more injected tokens are added. The results show a clear monotonic trend. While we acknowledge that the assumption is an empirical approximation rather than an absolute rule, these result provide evidence that the assumption is appropriate for harmful responses in practice.
>
> ---
>
> # W2. Complexity of RA Training
>
> As noted in the Limitations section, we acknowledge the high computational cost of RA training. At the same time, we would like to clarify that some of the cost, such as using a reward model, comes from RLHF itself and is not inherent to the design of RA. In addition, **generating intermediate states is inexpensive**, because we can simply create them by randomly selecting partial tokens from a set of pre-collected harmful responses. Overall, **the main contribution of RA is its training framework that accounts for corrupted intermediate states.** In future work, applying this framework to supervised learning may help address this limitation.
>
> ---
>
> # W3. Generality of Priming Token
>
> Thank you very much for your suggestions. We agree that it is valuable to discuss how more diverse tokens can “prime” the model. **We have strengthened the experiments in Appendix C.1 on how each token affects the model’s refusal probability.** Below, we summarize the experimental results.
>
> To dissect the effect of the inserted token, we consider three types of anchors:
>
> - **Affirmative tokens** include tokens such as “Yes”, “Sure”, and “Finally”, which explicitly suggest that the model is about to provide or continue an answer.
> - **Harmful tokens** consist of tokens such as “weapon” and “bomb” that are semantically aligned with the malicious intent of the query.
> - **Neutral tokens** include tokens such as “football” and “coffee”, which are semantically unrelated to the query and do not carry obvious safety connotations.
>
> Results (see Figure 4 in the revised version) show that **affirmative tokens reduce the refusal probability the most**. At the same time, **some neutral tokens such as “mountain” also decrease the refusal probability**. We speculate that this is because such words do not appear in typical refusal phrases during training and thus create an out-of-distribution pattern. In contrast, **harmful tokens do not consistently lower the refusal probability**, likely because they frequently appear within standard refusal justifications, and therefore remain compatible with high refusal probability.
>
> We plan to include these experiments in the main text of the camera-ready version if the page limit allows. Thank you again for your helpful suggestion.
>
> ---
>
> ### **Thank you again for taking the time to review our paper.**
>
> We are also truly grateful for your favorable evaluation of our work.

---

### Meta-Review · Area_Chair_Hf9m · 2026-01-09

**Summary:**

All reviewers agreed the paper makes a timely and meaningful contribution to DLM safety by identifying a priming vulnerability unique to diffusion-style decoding and by proposing a tailored mitigation via Recovery Alignment (RA). The strongest consensus strengths were the novelty of the vulnerability characterization, the practical and efficient First-Step GCG attack benchmark, and the empirical effectiveness of RA across multiple MDLMs/datasets. The main concerns raised were (i) the strength and clarity of the theoretical monotonicity assumption (R1, R4), (ii) the definition and generality of “affirmative tokens” and whether other token types can prime the model (R3, R1), (iii) computational cost and scalability of RA training (R1, R4), (iv) limited evaluation on larger-scale MDLMs (R3), and (v) presentation/related work completeness, including jailbreak evaluation metrics and prior priming work in ARMs (R2, R4). Overall, the committee judged the core empirical findings and defense contribution sufficiently strong and well-supported for acceptance despite some remaining limitations in theoretical generality and scaling discussion.

**Reviewer Concerns:**

The rebuttal and revision substantially addressed most key points. In particular, the authors clarified the role of the monotonicity assumption as an empirically motivated approximation and added supporting analyses in Appendix C, which resolved the main critique around theoretical validity (R1, R4). They also strengthened the discussion and experiments on token generality (affirmative vs. neutral vs. harmful) and clarified that affirmative tokens are not treated as a fixed vocabulary-wide set (R1, R3, R4). Concerns about related work coverage (metrics for jailbreak evaluation; priming vulnerability parallels in ARMs) were directly addressed via additions to the Related Work section (R2, R4), and the LLM usage disclosure and writing clarity issues were corrected (R4). Remaining outstanding issues are mostly limitations rather than blockers: RA training cost is acknowledged and still potentially significant, and evaluation on substantially larger MDLMs remains open due to lack of public models, but these do not undermine the main contribution given the current ecosystem.

**Reviewer Scores:**

Reviewer hNjZ (8) expressed strong overall support and would likely remain at 8 (possibly 8→9) after seeing added empirical validation and broader token analysis, though they still note computational cost. Reviewer zsC5 (8) had a single actionable concern (metrics-related work) that was directly addressed and would likely stay at 8. Reviewer ztRH (6) raised multiple clarity/generalization issues; given the clarified definition of affirmative tokens and added empirical validation, they would likely increase slightly to around 6→7, though they may still view scaling to larger MDLMs as an open limitation. Reviewer GBUt (6) explicitly stated that rebuttal experiments and clarifications addressed their concerns and that they might consider raising their score, so a reasonable estimate is 6→7 (or even 6→8 depending on weight placed on theory revisions), with their remaining feedback primarily about strengthening rigor rather than disputing the empirical contribution.

---

### Decision · Program_Chairs · 2026-01-26

Accept (Poster)